



**Modeling study of impacts on surface ozone of regional transport and**
**emission reductions over North China Plain in summer 2015**
Xiao Han[1,2], Lingyun Zhu[3], Shulan Wang[4], Xiaoyan Meng[5], Meigen Zhang[1,2], Jun Hu[4]
*1 State Key Laboratory of Atmospheric Boundary Layer Physics and Atmospheric Chemistry, Institute of*
*Atmospheric Physics, Chinese Academy of Sciences, Beijing 100029, China.*
*2 College of Earth Science, University of Chinese Academy of Sciences, Beijing 100049, China*
*3 Shanxi Province Institute of Meteorological Sciences, Taiyuan 030002, China*
*4 Chinese Research Academy of Environmental Sciences, Beijing, 100012, China*
*5 China National Environmental Monitoring Centre, Beijing, 100012, China*
Corresponding author:
Meigen Zhang
LAPC, Institute of Atmospheric Physics, Chinese Academy of Sciences
HuaYanBeiLi 40#, Chaoyang District
Beijing, China
Post code: 100029
Tel: 86-010-62379620
Fax: 86-010-62041393
E-mail: mgzhang@mail.iap.ac.cn
Lingyun Zhu
E-mail: mgzhang@mail.iap.ac.cn
Other authors:
Xiao Han, E-mail: hanxiao@mail.iap.ac.cn
Shulan Wang, E-mail: shulanwang@foxmail.com
Xiaoyan Meng, E-mail: mengxy@cnemc.cn
Jun Hu, E-mail: hujun@craes.org.cn



**Abstract**
Tropospheric ozone ($O_3$) has replaced $PM_{2.5}$ or $PM_{10}$ as the premier pollution in the North China Plain
(NCP) during summer in recent years. A comprehensive understanding of the $O_3$ production in responding
to the reduction of precursor emission over NCP is demanded urgently for the effective control policy
design. In this study, the air quality modeling system RAMS-CMAQ (regional atmospheric modeling
system-community multiscale air quality), coupled with the ISAM (integrated source apportionment
method) module is applied to investigate the $O_3$ regional transport and source contribution features during
a heavy $O_3$ pollution episode in June 2015 over NCP. The results show that the emission sources in
Shandong and Hebei were the major contributors to $O_3$ production in the NCP. Not only more than 50% $O_3$
mass burden in local regions, but also about 20-30% and 25-40% $O_3$ mass burdens in Beijing and Tianjin
were contributed by the emission sources in these two provinces, respectively. On the other hand, the urban
areas and most $O_3$ pollution regions of NCP were mainly dominated by the VOC-sensitive conditions, while
"both control" and $NO_x$-sensitive conditions dominated the suburban and remote areas, respectively. Then,
based on the sensitivity tests, the effects of several hypothetical scenarios of emission control on reducing
the $O_3$ pollution were compared and discussed. The results indicated that the emission control of industry
and residential sectors was the most efficient way if the emission reduction percentage was higher than
40%. However, when the emission reduction percentage dropped below 30%, the power plant sector could
make significant contributions to the decrease in $O_3$. The control strategies should be promptly adjusted
based on the emission reduction, and the modeling system can provide valuable information for precisely
choosing the emission sector combination to achieve better efficiency.













## 1. Introduction

In addition to the downward injection of stratospheric ozone ($O_3$), tropospheric $O_3$ is formed via a suite of photochemical reactions involving nitrogen oxides ($NO_x$), volatile organic compounds (VOCs), and sunlight. $O_3$ plays a role in controlling the chemical composition and climate of the troposphere and harms vegetation and human health, especially in industrialized regions (Kleinman et al., 2002). In recent years, the emission of $O_3$ precursors, $NO_x$ and VOCs, have increased substantially due to the economic growth, rapid population expansion, and urbanization in the North China Plain (NCP). During the summer, $PM_{2.5}$ or $PM_{10}$ are replaced by $O_3$ as the premier pollution type in major urban areas (China Environmental Status Bulletin 2015).

Numerous studies have investigated the spatial and temporal distribution characteristics of $O_3$ in the NCP. Lin et al. (2008) analyzed the three-year observation data of the $O_3$ mixing ratio at a remote Global Atmosphere Watch site near Beijing and showed the seasonal variation features of the $O_3$ background value for the NCP. Tang et al. (2012) gathered two-year observation data of the $O_3$ mixing ratio for 22 sites (located in urban, rural, and coastal areas) during a field campaign in the NCP and coupled the data with the meteorological parameters from the WRF. The spatial and temporal variations of $O_3$ were deeply analyzed, and the $O_3$–$NO_x$–VOCs sensitivity was initially investigated in this study. Ran et al. (2012) and Dufour et al. (2010) compared the $O_3$ seasonal variation features in megacities between the NCP and southern China. On the other hand, several studies have applied the chemistry transport model system to reproduce the three-dimensional $O_3$ continuous distribution characteristic and discussed the sensitivity of $O_3$ to precursor emissions (Wang et al., 2012; Nie et al., 2014).

Because of the strong emission of air pollutants, widespread haze clouds caused by serious air pollution have occurred frequently over the NCP (Tao et al., 2012; Wang et al., 2013; Li et al., 2016; Zhou et al., 2017). Aiming to solve this problem, the government has executed severe emission control strategies in recent years (Gao et al., 2016), which have yielded an initial effect. As reported by the China Environmental Status Bulletin, the mass loadings of sulfur dioxide ($SO_2$), $NO_2$, $PM_{2.5}$, and $PM_{10}$ steadily fell from 2013 to 2015. However, $O_3$ has become the only pollutant whose mass burden has continued to increase in the 74 experimental cities of China. The amount of surface $O_3$ is expected to continue increasing as the particulate mass loading decreases due to the emission control strategies employed in the NCP (Deng et al., 2011). Therefore, there is an urgent need to prevent environmental and health hazards in the NCP



resulting from the surface $O_3$.

As a secondary pollutant, although the basic features of surface $O_3$ in the NCP are well known from

measurement or modeling studies, understanding the chemical links between $O_3$ and its two main precursors,
$NO_x$ and VOC, is important for designing effective pollution reduction strategies (Castell et al., 2009). The
chemical transport model is an indispensable method for resolving the above issue, as it can quantify the
main physical and chemical mechanisms of pollutant formation and transport. Liu et al. (2010) used two
process analysis modules (integrated process rates and integrated reaction rates) embedded in CMAQ to
capture the dynamical and photochemical processes of $O_3$ formation in 2008 over China. As a result, the
influence and contribution of each important process can be distinguished and quantified. Tang et al. (2017)
also used the integrated process rates module for measurement data from a set of observation stations to
evaluate the sensitivity of $O_3$ production in June 2008 over the NCP. Xing et al. (2010) developed a
statistical response surface method and coupled it with CMAQ to analyze $O_3$ sensitivities to $NO_x$ and VOCs
emission changes in 2005 over eastern China. The overall impacts from individual sources, including
regional $NO_x$ and VOCs emission sources, have been evaluated using this modeling system. Li et al. (2008)
applied a tagged tracer method to the framework of NAQPMS to identify the transport contributions from
various $O_3$ production regions to total $O_3$ levels in 2008 over central eastern China. This method can be
used to eliminate the errors caused by nonlinearities in the transport and fast photochemistry of $O_3$ and its
precursors.

In general, the substantial features of $O_3$ formation sensitivity and the contributions of regional-scale

transport have been discussed in these studies. However, more work needs to be done to achieve a
comprehensive understanding of $O_3$ behavior over the NCP, especially the source contribution approaches
of recent years. In this study, an air quality modeling system called RAMS-CMAQ (regional atmospheric
modeling system–community multiscale air quality) that is coupled with the ISAM (integrated source
apportionment method) module is applied to estimate the regional contributions of $O_3$ among major regions
of the NCP and to quantify the relative amount of $O_3$ originating from specific VOCs and $NO_x$ emissions
sources. A unique method that can distinguish the $O_3$–$NO_x$–VOC sensitivity features is also used to identify
the precursor sensitivity regions and verify the results of the ISAM. In addition, the brute-force method is
applied to investigate the effect of reducing anthropogenic emissions on the $O_3$ mass burden. Therefore, the
precursor control type and contribution from specific geographic areas and emission sectors can be obtained,
and some valuable information can be provided for control strategies in the NCP.






### 2. Methodology

CMAQ (version 5.0.2), released in April 2014 by the US EPA
(https://www.airqualitymodeling.org/index.php?title=CMAQ_version_5.0.2_(April_2014_release)_Techn
ical_Documentation&oldid=587), was applied over the NCP for 2-month simulations in January and June
2015. Several updates and revisions, such as the chemical process corresponding to $NH_3$ and $SO_2$ and the
secondary aerosol formation of SOA (secondary organic aerosol) and nitrate, have been added in this
version. The updated and expanded version of the carbon bond mechanism (CB05) (Sarwar et al., 2008)
and the sixth-generation modal CMAQ aerosol model (AERO6) were applied to simulate the gas-phase
chemistry mechanisms and formation and the dynamic processes of aerosols, respectively. The
ISORROPIA model (version 1.7) was used to describe the thermodynamic equilibrium of gas-particle
transformation (Nenes et al., 1999). The highly versatile RAMS numerical code (Cotton et al., 2003), which
can well simulate the boundary layer and the underlying surface, is utilized to provide the meteorological
fields for CMAQ. The anthropogenic emissions of major pollution species ($NO_x$, $SO_2$, VOCs, BC, OC,
primary $PM_{2.5}$, and $PM_{10}$) were obtained from the monthly-based emission inventory, with 0.25°×0.25°
horizontal resolution and four categories (industry, power, transport, and residential), which were developed
to support the Model Intercomparison Study Asia (Li et al., 2015). The original version of this emission
inventory was developed for Asia as a contribution to the TRACE-P (Transport and Chemical Evolution
over the Pacific) Mission and ACE-Asia (Asian Pacific Regional Aerosol Characterization Experiment)
(Streets et al., 2003). Additionally, the $NO_x$ and $NH_3$ emissions from the soil and natural hydrocarbon
emissions were obtained from the Global Emissions Inventory Activity 1°×1° global monthly inventory
(Benkovitz et al., 1996). The Global Fire Emissions Database, Version 3 (FGEDv3.0; van der Werf et al.,
2010), was applied to provide the biomass burning emissions from wildfires, savanna burning, and slash-
and-burn agriculture. The VOCs and nitrogen oxides from flight exhaust, lighting, paint, fossil fuel, and
other sectors were obtained from the regional emission inventory in Asia (REAS, Version 2,
http://www.jamstec.go.jp/frsgc/research/d4/emission.htm) and the emission database for global
atmospheric research (Olivier et al., 1994), respectively.
The ISAM module was used to track $O_3$ from different geographic regions and source types. This
source apportionment tool was developed from the TSSA (Wang et al., 2009) in an early version of the
CMAQ model. Compared with the previous version, the ISAM improved the approach for the advection of



tagged tracers and the tracking of precursor reactions and increased the flexibility of the application by
minimizing the amount of data preparation (Kwok et al., 2013). An updated piecewise parabolic algorithm
was applied to reasonably estimate the major dynamics processes, including advection transport, vertical
diffusion, and dry deposition. For the nonlinear gas-phase chemical interactions, which are important for
$O_3$ formation, a hybrid approach that employs the direct sensitivity methods as linear equations using lower
and upper triangular matrices, which is known as LU decomposition (Yang et al., 1997), was applied for
description. In addition, the ISAM uses two tracers for individual nitrogen and VOC species to represent
the $O_3$ chemical formation regime attributed to either $NO_x$ or VOC emission sources. As described by Kwok
et al. (2014), the total concentration of $O_3$ in each model grid cell is equal to the sum of $O_3$ tracers that were
produced in either VOC- or $NO_x$-sensitive conditions:
$$O_{3\,bulk} = \sum_{tag} O_3 V_{tag} + \sum_{tag} O_3 N_{tag} \quad (1)$$
where $O_3V_{tag}$ and $O_3N_{tag}$ are the VOC-sensitive and $NO_x$-sensitive $O_3$ attributed to each tag source,
respectively. Therefore, the contribution from VOCs or $NO_x$ can be tracked individually, and the precursor
control types in each grid can be deduced. Detailed information regarding the ISAM can be found in Kwok
et al. (2013).

The simulation has two layer grids. The coarse domain covers East Asia (Figure 1, D1), with a

horizontal grid distance of 64 km and a total area of 6654 km×5440 km, and an inner domain (Figure 1,
D2) with a 16 km×16 km resolution is two-way nested with the outer one. The inner domain covers the
major regions of the NCP, including the megacity of Beijing, Tianjin, the capital city of the Shijiazhuang
province, Jinan, the industrial town of Tangshan, and the Hebei, Shandong, and Shanxi provinces. The
simulation used 15 vertical levels, of which nearly half were concentrated in the lowest 2 km, to improve
the simulation of the atmospheric boundary layer. Numerous previous studies have demonstrated that this
modeling system performs well in simulating the pollutant mass concentrations (Zhang et al., 2006; Han et
al., 2014; Han et al., 2016)

**3. Model evaluation**

The meteorological parameters, such as the temperature and wind field, are important impact factors

of ozone formation and transport. Therefore, the daily average temperature, relative humidity, wind speed,
and maximum wind direction in January and June 2015 were compared with the surface observation data
(released by the Chinese National Meteorological Center: http://data.cma.cn/) for Beijing, Tianjin,





Shijiazhuang, and Jinan. The comparison results are shown in Figure 2. The modeled temperature and
relative humidity are shown to generally coincide with the observations at all four of these stations, except
that some of the extreme high or low values appeared abruptly. The modeled wind speed, which could
reproduce the higher value in Tianjin and Jinan and lower value in Beijing and Shijiazhuang, also followed
the magnitude of observations well. However, a direct comparison between observed and modeled data is
difficult, especially for the wind vector, which can be easily influenced by the surrounding surface features.
The modeled and observed wind directions were not in good agreement with each other. Nevertheless, the
north wind in winter and south wind in summer were generally captured by the simulation results for all
stations.

The modeled mass concentrations of $O_3$ and one of its precursors ($NO_2$) were compared with the hourly

observation data from the Ministry of Environmental Protection of China; the results are shown in Figures
3 and 4. The statistical parameters – the means, standard deviation, and correlation coefficients between the
observations and simulations – are listed in Tables 1 and 2. The nitrogen oxide and tropospheric ozone were
two kinds of typical trace gases with high chemical activity and relatively short lifetimes. The diurnal
change in Figures 3 and 4 is obvious, and the distinctive values of the mass concentrations between different
seasons can also be found. The simulation results also reproduced these important features, especially the
evident diurnal variation of $O_3$ at these four stations. The mass concentration of $O_3$ in summer was generally
higher than that in winter because of the strong photochemistry during the daytime in summer. On the other
hand, the simulation results were able to capture most of the pollution episodes during these two months,
but the model still struggled to reflect some of the extreme high mass burden of the observations, such as
the high value points that appeared on June 22 and 25 in Beijing and on January 20 in Shijiazhuang. The
metrics listed in Tables 1 and 2 were used to evaluate the model performance, following the study of Yu et
al. (2006). Most of the correlation coefficients were higher than 0.5 for $NO_2$ and 0.6 for $O_3$, which indicates
that the model performed well in reproducing the observation trend. In addition, the standard deviations
between the observation and simulation of $NO_2$ and $O_3$ were also similar in most cases, except for $NO_2$ in
January. Figure 3 shows that the model missed many of the high values of observation that appeared during
the first half of January. In addition to the possible deficiency of the emission inventory, the model may not
adequately simulate the chemistry mechanism of nitrogen transformation during winter over the NCP.
However, this underestimation of $NO_2$ did not influence the performance of the $O_3$ simulation, as the mean
and standard deviation between the observation and simulation results were relatively close. The largest





212 deviation of the modeled $O_3$ mean, which was higher than that of the observation, appeared at Tianjin in

213 June, with the overestimation that occurred during June 16-18, as shown in Figure 3. Yu et al. (2010)

214 reported similar results and noted that the model might not well resolve the titration by NO in an urban area

215 under a low $O_3$ mass burden background by applying both the CB05 and SAPRC-99 mechanisms. ~~The~~

216 ~~slight overestimation could also be found at other stations.~~ Nevertheless, the comparison generally showed

217 that the model could basically reproduce the meteorological field and mass concentration and trends of $O_3$

218 and its precursor $NO_2$ during different seasons over the NCP.

220 **4. Results and discussion**

221 The surface spatial distributions of the monthly average values of the modeled $NO_x$, VOCs, and

222 maximum daily 8-hour average $O_3$ mass concentration (8H-$O_3$) for January and June 2015 are shown in

223 Figure 5. The monthly average wind field is also shown. The diffusion condition is shown to have been

224 weak due to the obviously smaller wind speed over Beijing, Tianjin, Hebei, Shandong, and northern Henan

225 in both January and June. In addition to the strong emission, this observation should be the main reason for

226 the high mass burden of $NO_x$ and VOCs in these regions. In addition, the maximum values were mainly

227 concentrated in the urban areas of the NCP during these two months, including the following five major

228 pollution cities: Beijing, Tianjin, Shijiazhuang, Jinan, and Tangshan. However, the distribution patterns

229 between $O_3$ and the precursors were significantly different, which indicates that the formation and transport

230 processes of $O_3$ should be complex in the NCP. Unlike the seasonal variation of $NO_x$ and VOCs, the mass

231 burden of $O_3$ in summer was obviously higher than that in winter because of the stronger photochemical

232 activity. The 8H-$O_3$ mass concentration, which exceeded the Grade II standard (160 μg m$^{-3}$), was

233 widespread throughout southern Beijing, Hebei and almost the entire areas of Tianjin and Shandong, with

234 values reaching 180-200 μg m$^{-3}$ in the tri-province area of Hebei, Shandong, and Henan in June. The serious

235 $O_3$ pollution was mainly concentrated in the northwest part of the Shandong province.

236 The contribution of $O_3$ from the major NCP regions, including Beijing, Hebei, Shandong, Tianjin, and

237 Shanxi, was calculated using ISAM-CMAQ-RAMS; the results are shown in Figure 6 (NS: $NO_x$-sensitive

238 $O_3$) and Figure 7 (VS: VOC-sensitive $O_3$). The total percentage can be obtained by summing the

239 contributors of NS and VS. The distribution patterns of NS and VS contributions were generally similar to

240 each other. The mass contribution of $O_3$ in Shandong, Hebei, and Shanxi was mainly contributed by local

241 sources, and the total percentage generally exceeded 50%. However, the local sources did not provide the





primary contributions in Beijing and Tianjin, and the regional contributions from Hebei and Shandong
could exceed 30% in these two cities, respectively. This feature clearly indicates that the regional transport
of precursors should be an important factor of $O_3$ pollution in Beijing and Tianjin. The contribution from
Shanxi to other regions was very small due to the hindrance to pollutant transport provided by the Taihang
Mountains, which are located to the east of the Shanxi province. In addition, the contribution from
Shandong provided at least more than 65% to the mass burden of $O_3$ in the Bohai Sea. This feature explains
the source of the large value that appears over this area in Figure 4. On the other hand, the contribution of
VS was obviously higher than that of NS in Beijing, Tianjin, Hebei, and Shandong. Compared with the NS,
the percentage of VS was generally double in Beijing and Tianjin and more than 10% higher in all of
Shandong and the southern part of Hebei. In contrast, the contribution of NS was clearly higher than that
of VS in Shanxi, which means that the major role of the $O_3$ formation in Shanxi should be different from
that in other regions.

To distinguish the $O_3$–$NO_x$–VOC sensitivity features, a method that is suitable for the results of three-

dimensional chemistry/transport models was applied to identify the precursor sensitivity regions in the NCP.
In addition to the base case, two sensitivity tests, which can reduce 30% of the VOC emissions or 30% of
the $NO_x$ emissions within the entire model domain, were conducted. Then, the deviation of the maximum
daily 8H-$O_3$ between the base case and these two sensitivity tests could be utilized to determine the
precursor control types in each grid. Here, we used $\Delta O_{3V}$ and $\Delta O_{3N}$ to represent the variation of the mass
concentration of $O_3$ due to the reduction in VOC or $NO_x$ emission, respectively (Sillman and West, 2009):
(1) if the changes in $\Delta O_{3V}$ and $\Delta O_{3N}$ were both less than 4 $\mu g\ m^{-3}$, this grid was likely controlled by neither
$NO_x$ nor VOCs; (2) if $\Delta O_{3N}$ increased to a value greater than 4 $\mu g\ m^{-3}$ and $\Delta O_{3V}$ decreased to a value less
than 4 $\mu g\ m^{-3}$, this grid should be regarded as "$NO_x$ titration"; (3) if $\Delta O_{3V}$ decreased by more than 4 $\mu g\ m^{-3}$
, with this reduction being twice as large as the $\Delta O_{3N}$ reduction (or $\Delta O_{3N}$ increase), this grid was likely
controlled by VOCs; (4) if $\Delta O_{3N}$ decreased by more than 4 $\mu g\ m^{-3}$, with the reduction being twice as large
as the $\Delta O_{3V}$ reduction, this grid was likely controlled by $NO_x$; (5) if $\Delta O_{3N}$ and $\Delta O_{3V}$ both decreased by more
than 4 $\mu g\ m^{-3}$ and the ratio between them was less than 2:1 or 1:2, this grid was likely controlled by both
$NO_x$ and VOCs. Details regarding the identification explained above can be found in Figure 7(f). The
frequency of precursor control types in each grid in June was determined and is shown in Figure 8(a-e).
The $NO_x$ titration scarcely appeared in the model domain. The frequency of the "no control" type entirely
exceeded 50% over the background regions when the $O_3$ mass burden was lower than 120 $\mu g\ m^{-3}$ and



gradually decreased as the $O_3$ mass burden increased. Over the $O_3$ pollution areas, a grid with a "no
control"-type frequency higher than 10% was seldom found. Specific to the considered regions, the urban
area of Beijing, Tianjin, Tangshan, southern Hebei, and northern and western Shandong were mainly under
VOC control, while the outer suburb of Beijing, all of Shanxi, and northern Hebei were mainly under $NO_x$
control. The "both control" type generally appeared in the transitional zone between $NO_x$ and VOC control.
Compared with the results shown in Figures 6 and 7, the distribution feature of $NO_x$ and VOC contributions
highly coincided with that of the $O_3$ precursor sensitivity types, which demonstrated that this method is
reliable.
In addition to the contribution feature of emission sources estimated using the ISAM, the effect of
reducing anthropogenic emissions on the $O_3$ mass burden was also necessary to learn because the formation
of $O_3$ from $NO_x$ and VOC emissions is a typical nonlinear process. The brute-force method, which can
realistically capture the nonlinear processes of secondary pollutant formation, was applied. Therefore,
several sensitivity tests were designed, as shown in Table 3. First, the zero-out (100% source removal)
simulations of four major sectors, i.e., industry, power plants, transport, and residential (sensitivity tests ZI,
ZP, ZT, and ZR, respectively), were conducted to evaluate the efficiency of emission reduction for different
sources in the NCP. Figure 9 presents the results of the brute-force sensitivity tests and the $NO_x$ and VOC
emission flux of each major sector. The removal of the industry sector is shown to have been the most
efficient way to decrease the $O_3$ mass burden, and the variation of 8H-$O_3$ between 20 and 30 μg m$^{-3}$ was
generally concentrated in the high mass concentration regions. The main reason is likely that the VOC
emission flux of the industry sector was significantly higher than that of the other sectors. Removal of the
residential sector could also decrease the $O_3$ mass burden in most of the VOC control regions due to its
VOC emission flux being notably higher than that of $NO_x$. In contrast, removal of the transport and power
plant sectors could not effectively reduce the $O_3$ mass burden and even increased the mass burden in high
pollution areas, such as southern Beijing, Tianjin, Tangshan, southern Hebei, Jinan, and other parts of
Shandong. The $NO_x$ emission flux of these two kinds of sectors was clearly higher than that of VOCs,
especially for the power plant sector. It also caused the 8H-$O_3$ mass burden to decrease by 5-10 μg m$^{-3}$ in
Shanxi as a result of the removal of the power plant sector. In summary, if we focus on the major pollution
regions of the NCP, including Beijing, Tianjin, Hebei, and Shandong, reduction of the industry and
residential emission sectors should be an effective way to control the $O_3$ mass burden during heavy $O_3$
pollution episodes.



In addition, the realistic pollution control strategies are supposed to be applied to a specific sector in
the high emission regions (HERs) and used to develop a comprehensive reduction scheme; thus, a detailed
analysis is necessary to investigate more accurate and practical strategies. Other than applying the simple
zero-out sensitivity test over entire objective regions, we selected the regions that include cities and towns
with high anthropogenic emission flux in the Beijing, Tianjin, Hebei, and Shandong (BTHS) region to more
accurately match real emission control. Figure 10 presents the selected regions and the emission flux of
$NO_x$ and VOCs from the industry sector, residential sector, and multiple combinations. First, the change in
8H-$O_3$ mass concentration associated with the anthropogenic emission in selected regions (Figures 10(i)
and 10(j)) was compared with that in the entire BTHS region (sensitivity tests A20%-HERs and A20%-
BHTS), as shown in Figures 11(a) and 11(b), respectively. The distribution patterns of the 8H-$O_3$ mass
burden variation were notably similar to each other, and the positive and negative values generally appeared
in the same regions. However, the negative value in Figure 11(b) was clearly higher than that in Figure
11(a). This disparity indicates that significant overestimation of the $O_3$ mass burden variation might occur
when we conduct a brute-force sensitivity test with broad reductions in emissions in the entire objective
regions.
According to the results of the zero-out sensitivity tests, the industry and residential sectors were the
major emission sources of $O_3$, while the power plant sector did not benefit $O_3$ formation. Thus, the effects
of reducing these industry and residential sectors were estimated using the brute-force method with 20%
emission intensity in the selected regions of BTHS (Figures 10(a) and 10(b)). Figures 11(c) and 11(d) show
the variation of $O_3$ associated with the industry and residential emission sectors (sensitivity tests I20%-
HER and R20%-HER), respectively. The 8H-$O_3$ mass concentration could decrease by 10-12 μg m$^{-3}$ in
most of Shandong, especially in the strong polluted regions shown in Figure 11(c). In contrast, the value
slightly increased in the urban areas of Shijiazhuang, Tianjin, and Tangshan. In Figure 5(f), the 8H-$O_3$ mass
burden was relatively lower in these regions. Thus, the $O_3$ mass burden can be decreased rapidly by
controlling the industry emissions under a heavy $O_3$ pollution background. Figure 11(d) shows that the 8H-
$O_3$ mass concentration decreased overall in BTHS, though the range was only 1-5 μg m$^{-3}$. The likely main
reason is that the emission of VOCs was higher than that of $NO_x$ from the residential sector, while the
emission intensity from the residential sector was relatively lower than that from industry. The mass burden
of $O_3$ can also be reduced by controlling the residential emissions in the urban areas of Shijiazhuang, Tianjin,
and Tangshan.





In addition, the influence of different combinations of emission sectors in BTHS was discussed.
Figures 11(e) and 11(f) present the change in 8H-$O_3$ mass concentration associated with a 20% emission
intensity for both the industry and residential sectors (sensitivity test IR20%-HERs) and the industry,
transport, and residential sectors (sensitivity test ITR20%-HERs), respectively. The $O_3$ mass burden
generally decreased sharply in BTHS, as shown in Figure 11(e), especially in the regions of Shandong with
heavy pollution. The range and magnitude of decrease can obviously be enhanced while considering the
reduction of the transport sector, as shown in Figure 11(f). Notably, the mass concentration of 8H-$O_3$ could
decrease from 180-200 $\mu g\ m^{-3}$ to 160-180 $\mu g\ m^{-3}$ in the polluted regions of BTHS. Compared with the zero-
out sensitivity test in Figure 9, the decrease in 8H-$O_3$ mass burden in Figure 11(f) was still clearly lower
than that of ZI. This deviation indicates that the contribution source from other regions except BTHS should
also be important. Even though 80% of the emission intensity was removed, the reduction in 8H-$O_3$ mass
concentration still barely exceeded 20 $\mu g\ m^{-3}$ in the NCP, as shown in Figures 11(c), 11(d), and 11(e), which
means that it was difficult to keep the $O_3$ mass burden under the Grade II standard by controlling only the
industry and residential emission sectors in HERs.
Therefore, more brute-force sensitivity tests with HERs emissions varied from 50% to 0% were
conducted. The regional average 8H-$O_3$ mass concentrations in Beijing, Tianjin, Shijiazhuang, Jinan, and
Tangshan with changes in emission are shown in Figure 12. Three series of sensitivity tests were conducted:
reduction of the IR (industry and residential), ITR (industry, transport, and residential) and All (industry,
transport, power plant, and residential) emission sectors. As shown, the 8H-$O_3$ mass concentration was
higher than 160 $\mu g\ m^{-3}$ in all five cities, while the emission percentage was 100%. When the emissions
reduced to 50%, the 8H-$O_3$ mass concentrations of these three series slightly decreased for Beijing, Tianjin,
Tangshan, and Jinan but increased for Shijiazhuang. The decrease in 8H-$O_3$ mass concentration as a result
of reducing the IR emission was similar to that of the ITR emission when the emissions were reduced from
50% to 40% for all five cities but was not significant when the reduction was less than 40%. The lines
corresponding to the ITR and All emission sectors generally decreased coherently for these five cities when
the emissions were reduced from 50% to 30%. However, the effect of the ITR reduction was obviously
weaker than that of the All reduction when the reduction was less than 30%. The decrease in 8H-$O_3$ mass
burden exceeded 12 $\mu g\ m^{-3}$ when the All emission reduction was least, and the air quality in all five of these
cities could reach the Grade II standard. This phenomenon indicated that the influence of the transport and
power plant emission sectors on the decrease in $O_3$ mainly occurred after removing 60% of the IR or 70%



of the ITR emission intensity, respectively. Thus, an emission control sequence for different sectors should
be considered when exploring more effective strategies.

**5. Conclusions**

In this study, an air quality modeling system referred to as RAMS-CMAQ was applied to simulate the

$O_3$ mass concentration, and several sensitivity tests were conducted to investigate the $O_3$ pollution and to
discuss the relationship between $O_3$ production and emission contributions over the NCP in January and
June of 2015. First, the modeled daily meteorological factors (temperature, relative humidity, and wind
field) and hourly mass concentrations of $O_3$ and its precursor $NO_2$ were compared with ground-based
observation data to evaluate the accuracy and reliability. The simulation results were generally good and
able to broadly capture the values and variation trend of the observation data. Focusing on the heavy $O_3$
pollution period in June, an advanced source apportionment tool called ISAM was coupled with RAMS-
CMAQ and applied to estimate the regional transport contributions, with individual tracers for nitrogen and
VOC species used to represent the $O_3$ chemical formation regime attributed to either $NO_x$ or VOC emission
sources in the NCP. Then, a unique method that is suitable for three-dimensional chemistry/transport models
was used to distinguish the $O_3$–$NO_x$–VOC sensitivity features and identify the precursor sensitivity in each
grid of the model domain. Therefore, the $O_3$ mass burden sensitivities to $NO_x$ and VOC emission changes
and the correlative regional transport contribution features among major anthropogenic source regions in
the NCP can be clearly investigated using these methods. In addition, several brute-force sensitivity tests
were conducted to discuss the role of the main anthropogenic emission sectors on reducing the $O_3$ mass
burden, and an attempt was made to provide valuable suggestions for exploring more effective strategies
for preventing $O_3$ pollution. The results are summarized as follows:

1. The simulation results show that the seasonal variation of $O_3$ was significant and that the heavy

mass burden of 8H-$O_3$, which exceeded the Grade II standard, generally occurred in southern Beijing, Hebei
and almost all of Tianjin and Shandong in June. The mass burden of 8H-$O_3$ reached 180-200 μg m$^{-3}$ mainly
in the tri-province area of Hebei, Shandong, and Henan. The distribution pattern and seasonal variation of
8H-$O_3$ were obviously different from those of its precursors, which indicates that the formation and
transport processes of $O_3$ should be complex in the NCP.

2. The results of RAMS-CMAQ-ISAM show that the emission sources in Shandong and Hebei were

the major contributors to $O_3$ production in the NCP. In addition to these two provinces, the $O_3$ mass burden





in Beijing and Tianjin was also significant. The emissions from Hebei and Shandong contributed 15-20%
and 5-10% to Beijing and 10-20% and 15-20% to Tianjin, respectively. However, the $O_3$ mass burden in
these two provinces was generally contributed by the provinces themselves. The results also show that the
contribution of VS was clearly higher than that of NS in Beijing, Tianjin, Hebei, and Shandong, which
indicates that the $O_3$ mainly originated from VOC emission sources. On the other hand, the emission sources
in the Shanxi province almost had no impact on the $O_3$ mass burden in other regions of the NCP due to the
hinderance to pollutant transport provided by the Taihang Mountains.

3. The results of identification of the $O_3$–$NO_x$–VOC sensitivity feature show that the VOC control

mainly occurred over all of Tianjin and Tangshan and southern Beijing (urban area) and Hebei, where the
$O_3$ mass concentration reached 160-180 μg m$^{-3}$. The north central part of Shandong and urban area of Jinan
were also mainly under the VOC control. The frequencies of VOC control and the "both control" type was
generally equal in the region of Hebei and Shandong where the $O_3$ mass concentration reached 180-200 μg
m$^{-3}$. The $NO_x$ control generally appeared in the regions of the NCP where the $O_3$ mass concentration reached
120-160 μg m$^{-3}$. In the major cities with $O_3$ pollution, including Beijing, Tianjin, Shijiazhuang, and Jinan,
the $O_3$–$NO_x$–VOC sensitivity feature was the same: VOC control dominated the urban area, while "both
control" and $NO_x$ control dominated the suburban and remote areas, respectively.

4. The results of the zero-out sensitivity tests show that the IR emission sectors were two important

contributors to ozone formation, as they were the major sources of VOCs, while the power plant emission
sector did not benefit $O_3$ pollution control in the high mass burden regions due to the greater emission of
$NO_x$ versus VOCs.

On the other hand, the results of the brute-force sensitivity tests show that the effects of IR, ITR, and

All emission control on the decrease in $O_3$ were similar when their reduction percentages were higher than
40%. Meanwhile, the effects of ITR and All emission control were similar while the reduction percentages
were higher than 30%. When the reduction percentage dropped below 30%, the nonlinearity of $O_3$
formation was notable, and the power plant sector could make significant contributions to the decrease in
$O_3$. Thus, the control strategies should be promptly adjusted based on the emission reduction, and the
emission sector combination should be precisely chosen to achieve better efficiency. The modeling system
allows us to capture valuable information regarding how to choose the correct sequence and efficient
combinations by exploring the key thresholds from the bulk of sensitivity tests regarding crucial parameters.





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




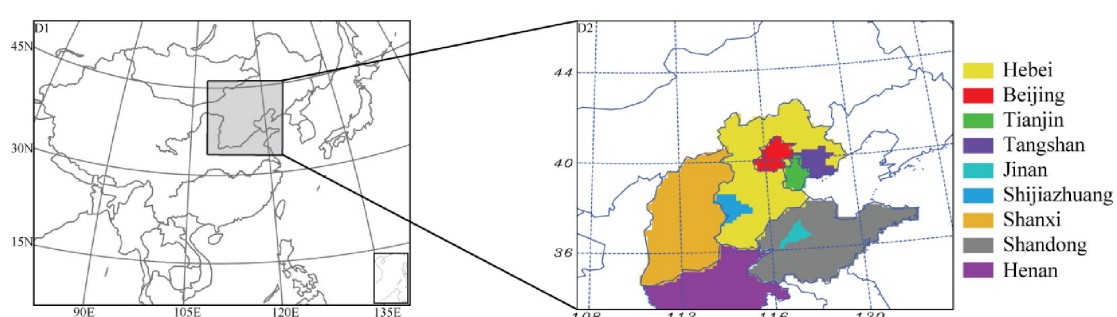

Figure 1 The model domain of this study, and the geographic locations of Beijing, Tianjin, Tangshan, Hebei,
Shijiazhuang, Shanxi, Shandong, Jinan, and Henan.




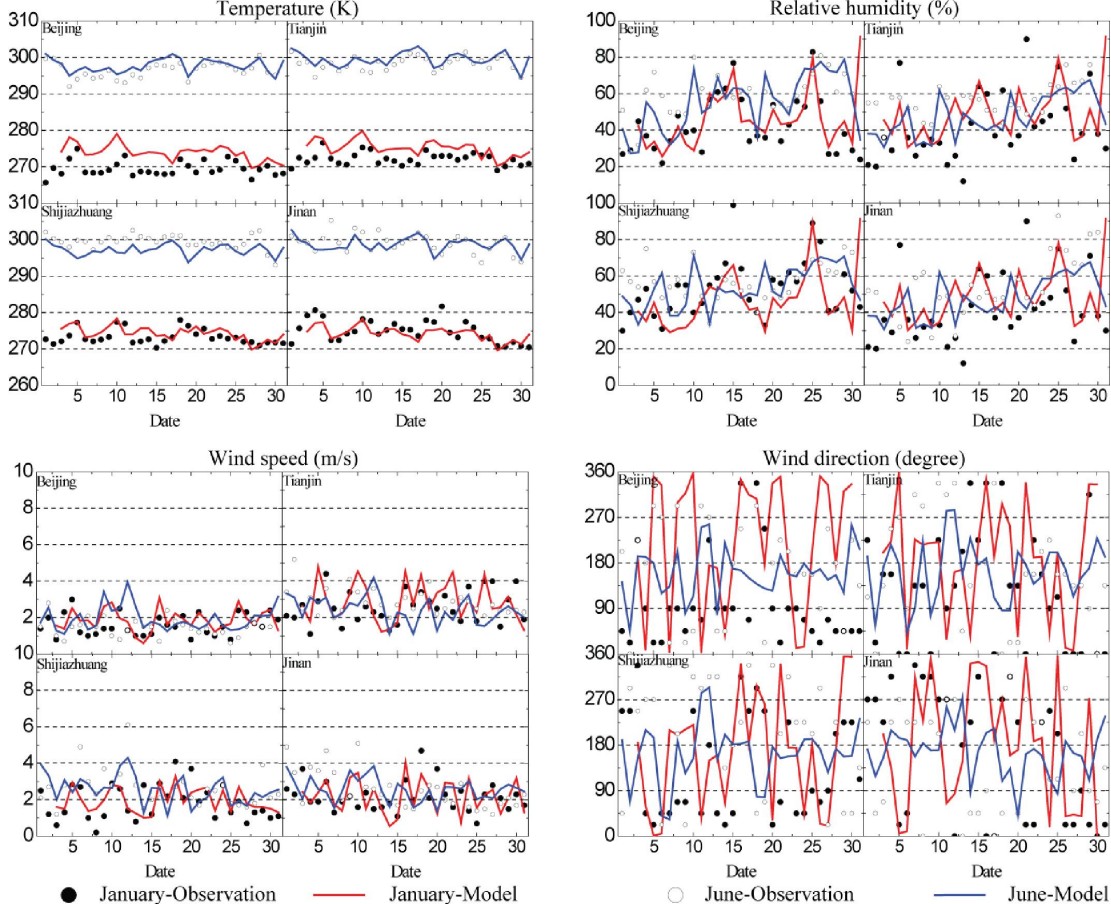


Figure 2.Observed and modeled daily average temperatures (K), relative humidity (%), wind speed (m/s), and maximum
wind direction at four stations in January and June 2015.






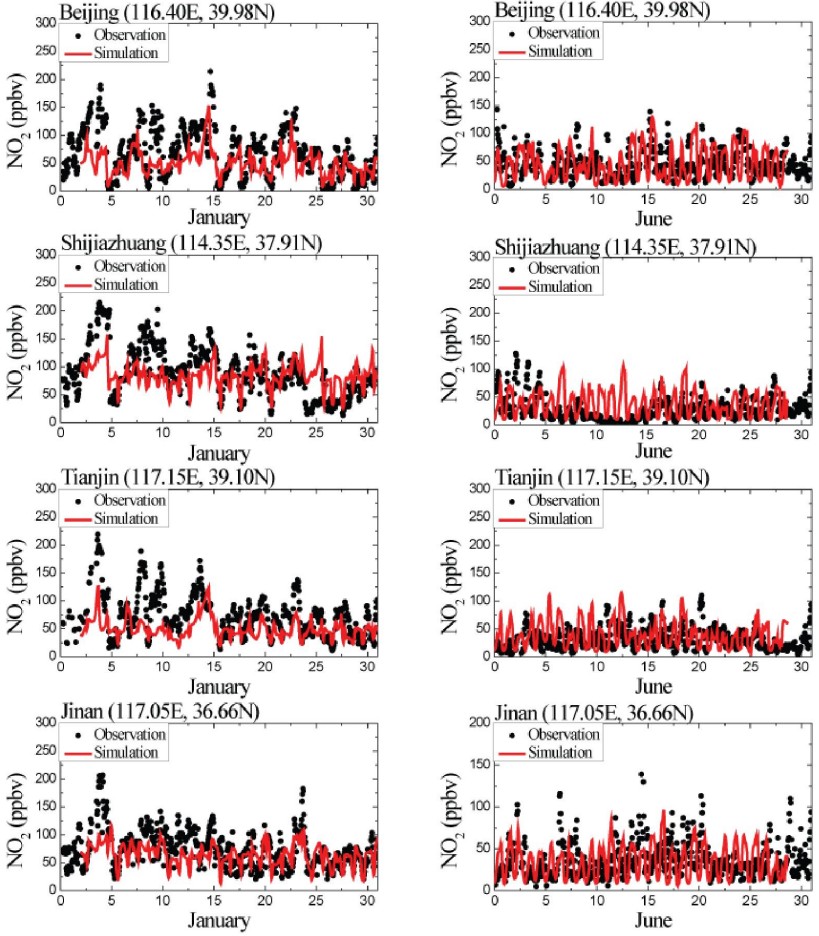

Figure 3. Observed (black circles) and modeled (red solid lines) hourly mass concentrations (μg m$^{-3}$) of NO$_2$ at Beijing,
Shijiazhuang, Tianjin, and Jinan in January and June 2015.





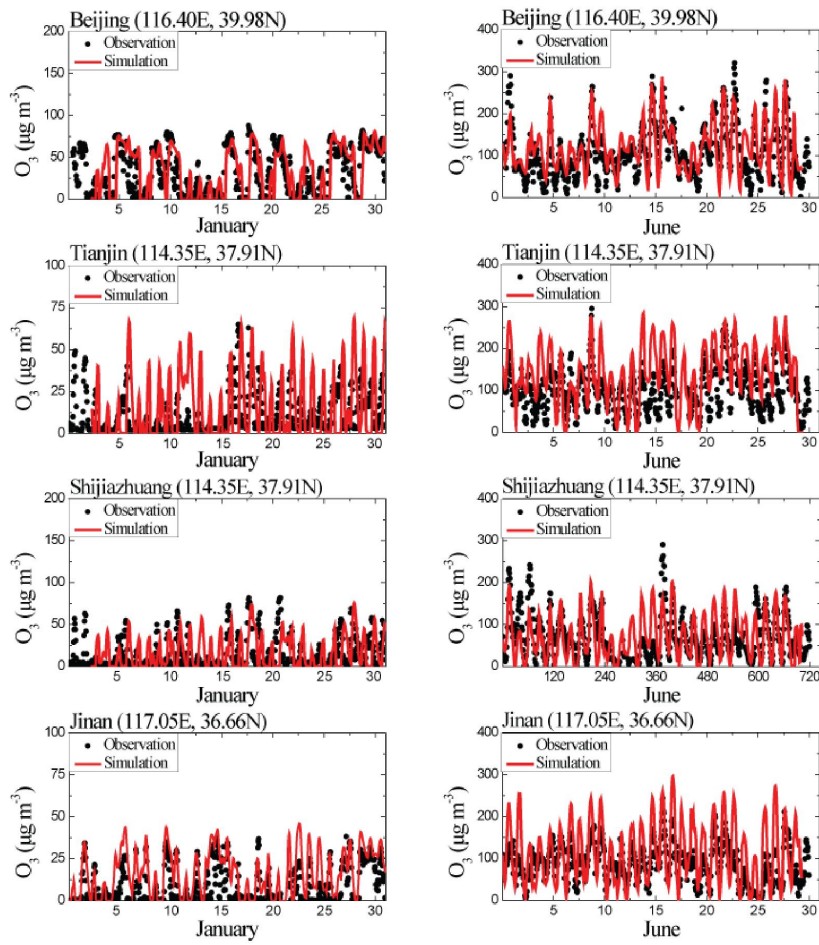


Figure 4. Observed (black circles) and modeled (red solid lines) hourly mass concentrations (µg m⁻³) of O₃ at Beijing,


Shijiazhuang, Tianjin, and Jinan in January and June 2015.


















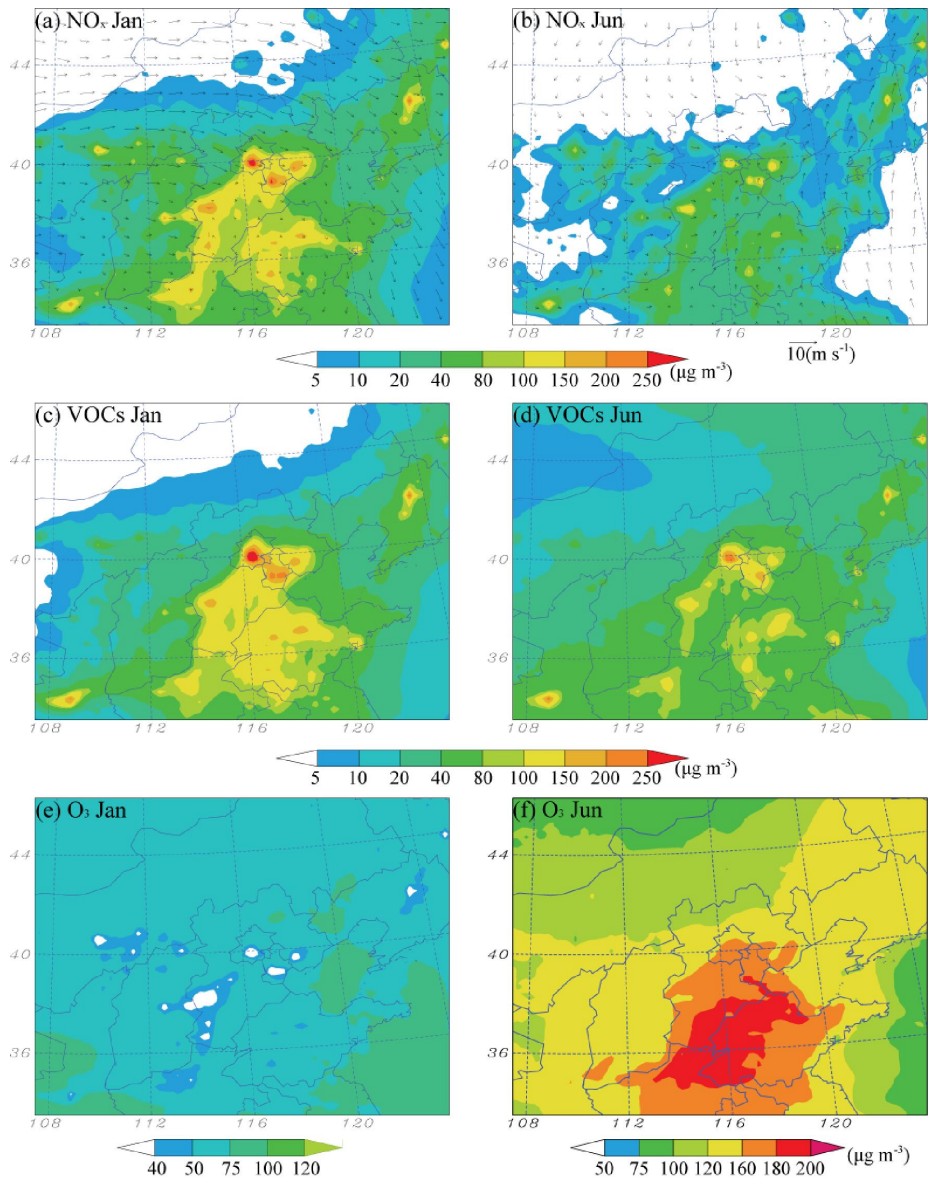

Figure 5. The surface spatial distributions of monthly average NO$_x$ (a-b) and VOCs (c-d), and maximum daily 8H-O$_3$ (e-
f) in January and June 2015.





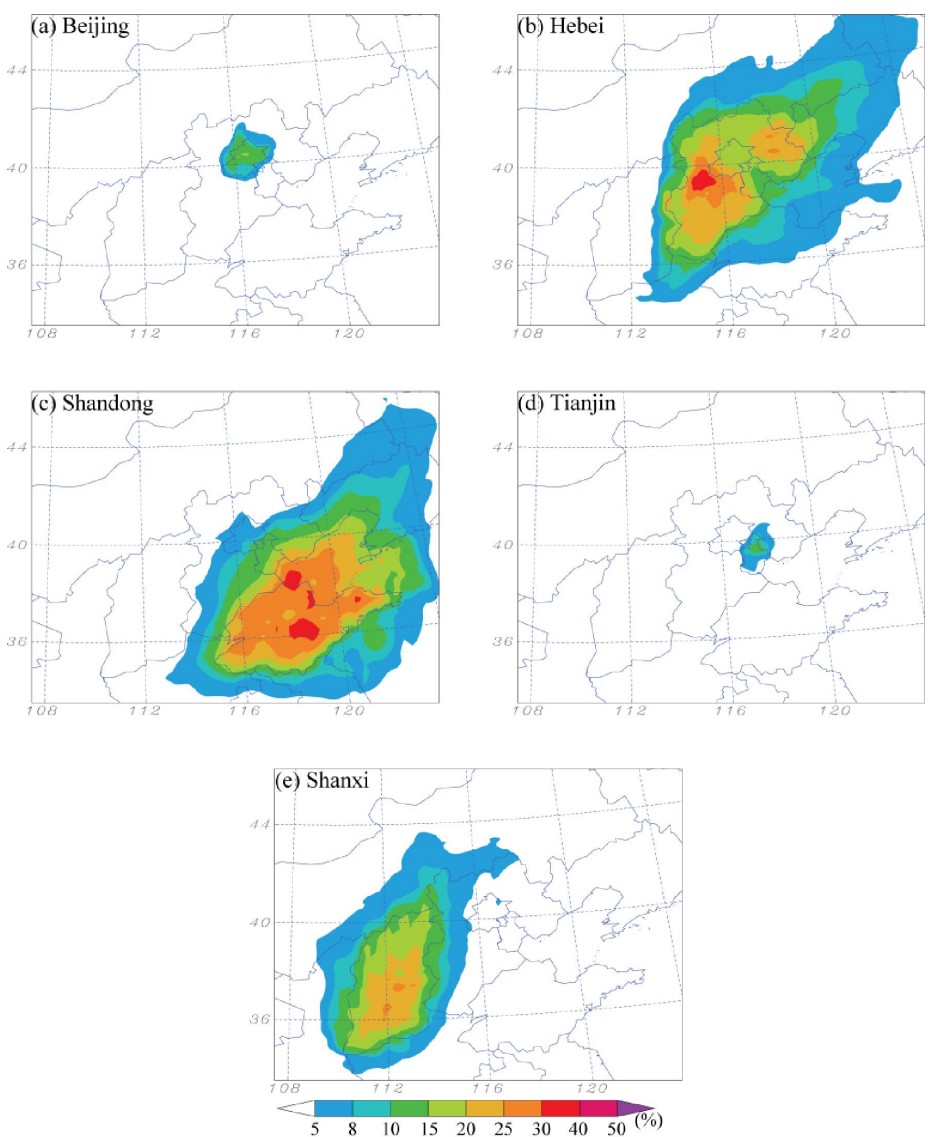


Figure 6. The regional contribution of NO$_x$-sensitive O$_3$ from (a) Beijing, (b) Hebei, (c) Shandong, (d) Tianjin, and (e)
Shanxi in June 2015.





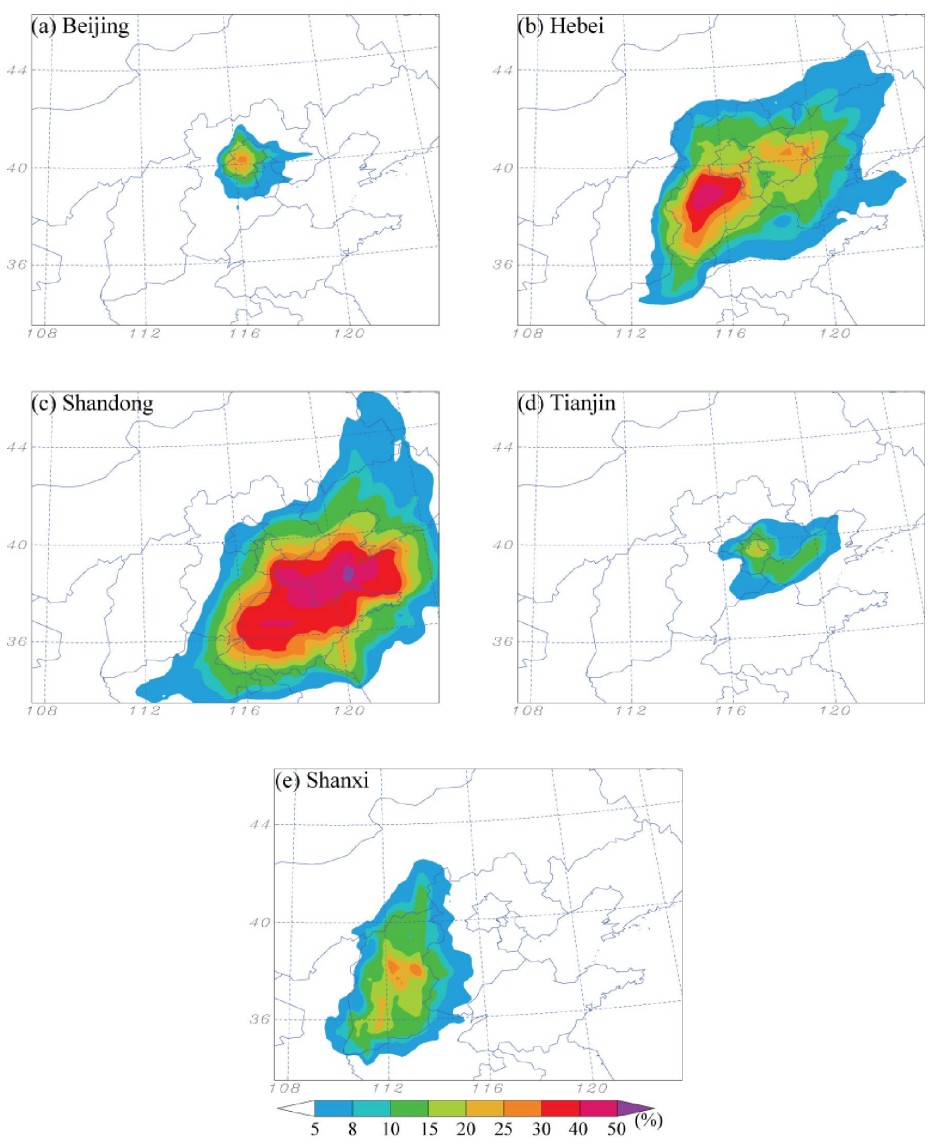

Figure 7.Same as Figure 5 but for VOC-sensitive $O_3$.





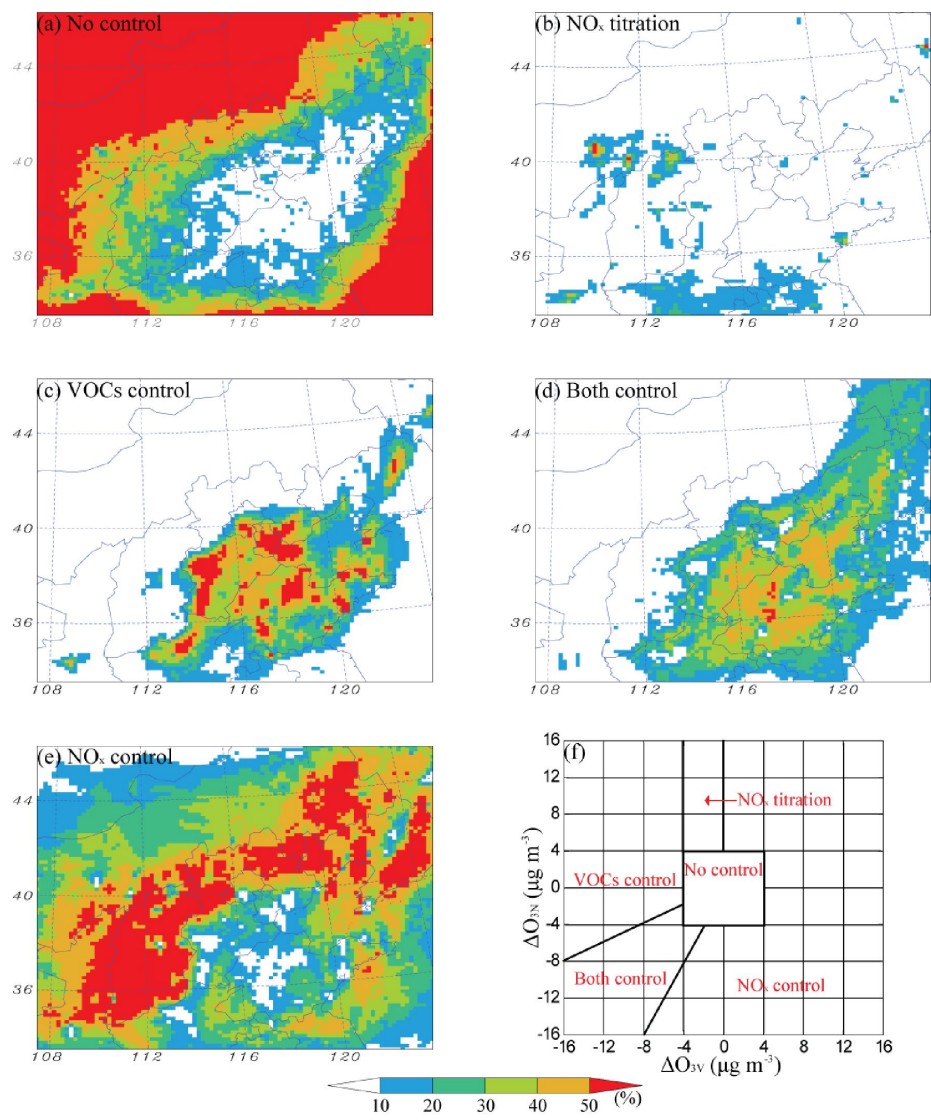

Figure 8. Distributions of the frequency of 8-hour ozone precursor sensitivity regions in June 2015.





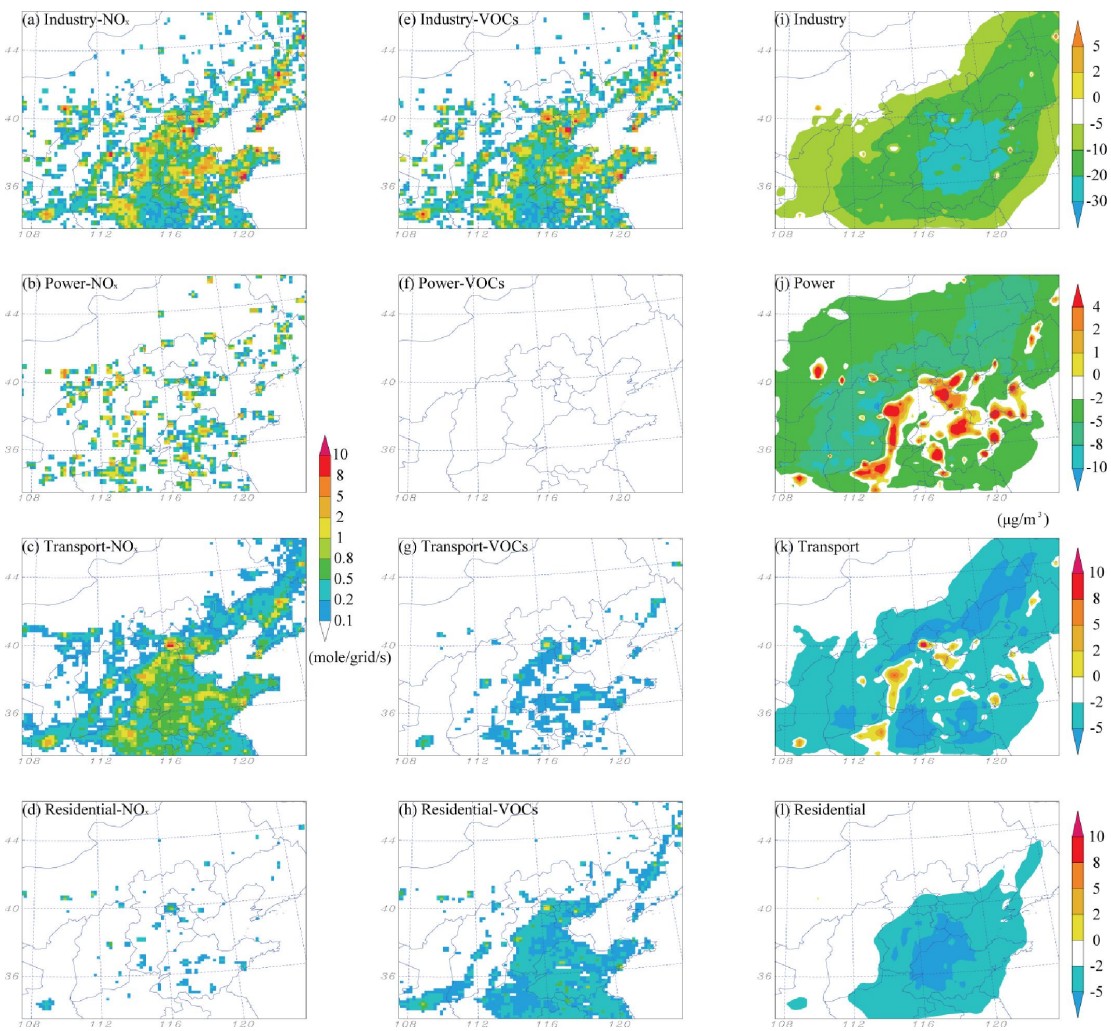


Figure 9. Distributions of the emission flux of NOx and VOCs and the variation of mass concentration of 8H-O₃ associated with the ZI, ZP, ZT, and ZR in June.


















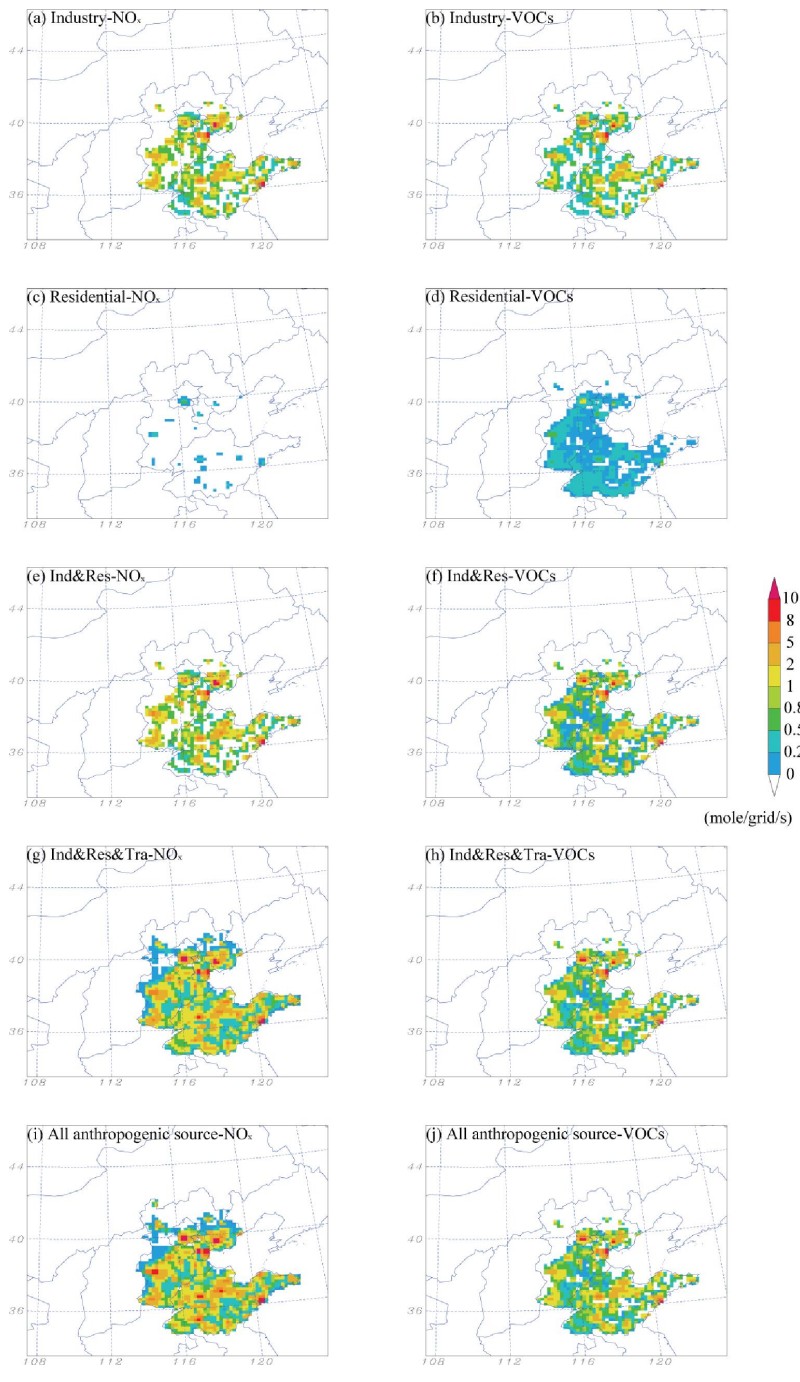


Figure 10. Distributions of the $NO_x$ and VOCs emission flux from different sectors or combinations in the high emission regions of Beijing, Tianjin, Hebei, Shandong in June.




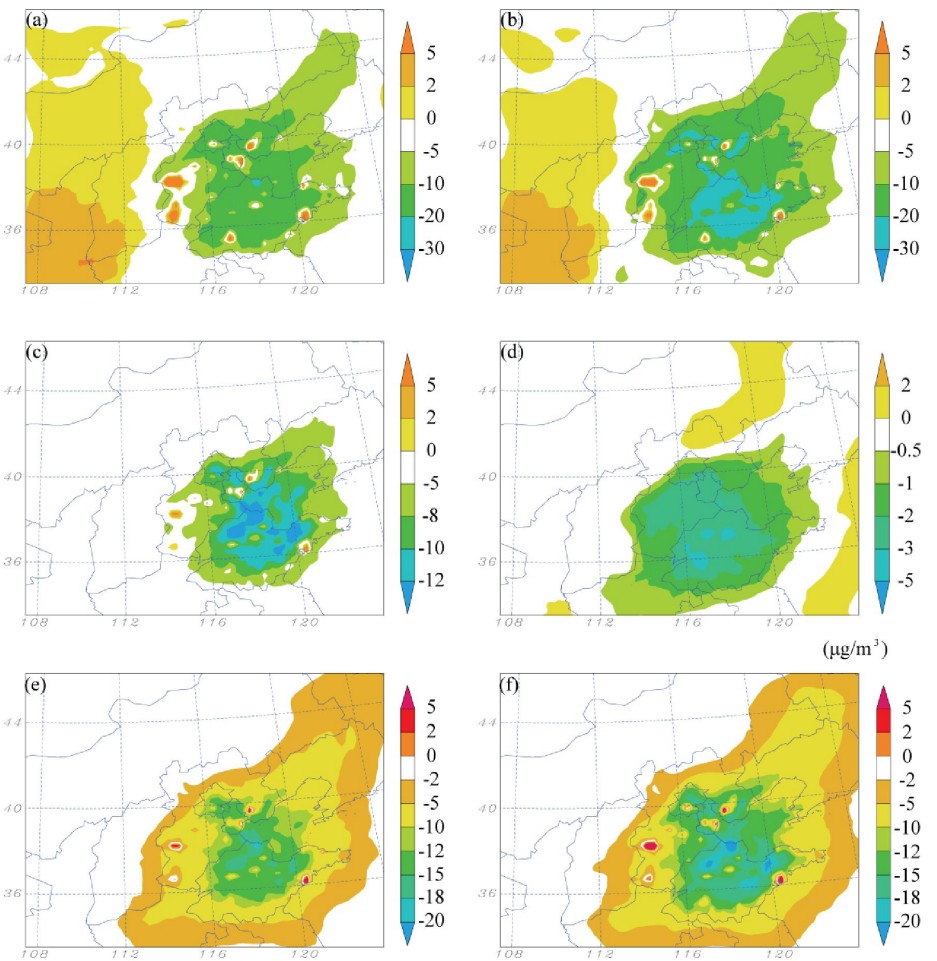


Figure 11. Distributions of the variation of 8H-O$_3$ mass concentration associated with brute force sensitivity tests: (a)

A20%-HER; (b) A20%-BHTS; (c) I20%-HER; (d) R20%-HER; (e) IR20%-HER; (f) ITR20%-HER.
















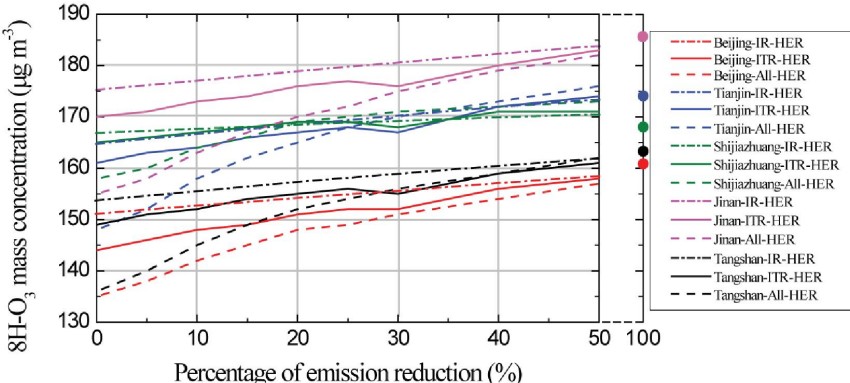


Figure 12. The variation of regional average 8H-O₃ mass concentrations in Beijing, Tianjin, Shijiazhuang, Jinan, and
Tangshan with reduction of IR, ITR and All emissions, respectively.





Table 1. Statistical summary of the comparisons of the hourly NO$_2$ comparison between simulation and observation

|       |              |     | $N^a$ | $C_{obs}^b$ | $C_{mod}^c$ | $\sigma_{obs}^d$ | $\sigma_{mod}^e$ | $R^f$ |
|-------|--------------|-----|-------|-------------|-------------|--------------------|--------------------|-------|
|       | Beijing      | Jan | 602   | 68.71       | 50.62       | 42.98              | 21.7               | 0.59  |
|       |              | Jun | 588   | 45.39       | 46.75       | 24.95              | 28.49              | 0.53  |
|       | Jinan        | Jan | 616   | 74.39       | 63.26       | 33.98              | 19.55              | 0.55  |
| NO$_2$ |              | Jun | 639   | 34.27       | 34.93       | 19.57              | 18.76              | 0.47  |
|       | Shijiazhuang | Jan | 618   | 90.04       | 83.78       | 44.91              | 21.55              | 0.54  |
|       |              | Jun | 629   | 26.11       | 38.82       | 21.59              | 22.26              | 0.44  |
|       | Tianjin      | Jan | 584   | 73.73       | 49.07       | 37.94              | 18.52              | 0.55  |
|       |              | Jun | 639   | 30.02       | 40.29       | 18.36              | 23.25              | 0.52  |

[a] Number of samples

[b] Total mean of observation

[c] Total mean of simulation

[d] Standard deviation of observation

[e] Standard deviation of simulation

[f] Correlation coefficient between daily observation and simulation































Table 2. Statistical summary of the comparisons of the hourly $O_3$ comparison between simulation and observation

| | | | N | $C_{obs}$ | $C_{mod}$ | $\sigma_{obs}$ | $\sigma_{mod}$ | R |
|---|---|---|---|---|---|---|---|---|
| $O_3$ | Beijing | Jan | 615 | 33.57 | 37.88 | 27.58 | 27.2 | 0.54 |
| | | Jun | 676 | 106.96 | 120.85 | 63.75 | 57.33 | 0.74 |
| | Jinan | Jan | 673 | 11.09 | 13.58 | 10.75 | 13.08 | 0.74 |
| | | Jun | 693 | 87.91 | 111.44 | 45.54 | 71.8 | 0.62 |
| | Shijiazhuang | Jan | 627 | 15.24 | 18.54 | 18.74 | 18.7 | 0.57 |
| | | Jun | 692 | 69.53 | 71.78 | 53.15 | 76.14 | 0.65 |
| | Tianjin | Jan | 629 | 10.83 | 17.05 | 11.78 | 19.36 | 0.48 |
| | | Jun | 675 | 100.42 | 143.31 | 52.22 | 69.48 | 0.74 |





Table 3. The brute force sensitivity tests set in this study

|   | Abbreviation | Brute force sensitivity test |
|---|---|---|
| 1 | ZI | Zero-out of industry emission sector |
| 2 | ZP | Zero-out of power plants emission sector |
| 3 | ZT | Zero-out of transport emission sector |
| 4 | ZR | Zero-out of residential emission sector |
| 5 | A20%-BHTS | 20% emission of all anthropogenic sectors in BTHS |
| 6 | A20%-HER | 20% emission of all anthropogenic sectors in the selected high emission regions of BTHS |
| 7 | I20%-HER | 20% emission of industry sector in the selected high emission regions of BTHS |
| 8 | R20%-HER | 20% emission of residential sector in the selected high emission regions of BTHS |
| 9 | IR20%-HER | 20% emission of industry and residential sector in the selected high emission regions of BTHS |
| 10 | ITR20%-HER | 20% emission of industry, transport, and residential sector in the selected high emission regions of BTHS |
