# Peer review of "Modeling study of impacts on surface ozone of regional transport and emission reductions over North China Plain in summer 2015"

_Atmospheric Chemistry and Physics, 2018_

## Referee Comment (RC1) · Anonymous Referee #2 · 21 May 2018

In the manuscript, the air quality modeling system RAMS-CMAQ (regional atmospheric modeling system-community multiscale air quality), coupled with the ISAM (integrated source apportionment method) module is applied to investigate the O3 regional transport and source contribution features during a heavy O3 pollution episode in June 2015 over NCP. It explores that that the emission sources in Shandong and Hebei was the major contributors to O3 production in the NCP, and it found that the modeling system can provide valuable information for precisely choosing the emission sector combination to achieve better efficiency. It is meaningful. I recommend the manuscript to be accepted after some minor revisions, and detail some issues below.

Major points:

1. The modeled and observed wind directions were not in good agreement with each other, even in Jan. and Jun. How can you get the result that about 20-30% and 25-40% O3 mass burdens in Beijing and Tianjin were contributed by the emission sources in Shandong and Hebei? Whether should the author compare with the regional atmospheric circulation field?

2. In Figure 4, it seems that there are negative values in modeled hourly mass concentrations of O3 in January, how does this result happen?

3. In Figure 3, the model doesn't perform well in reproducing the observation trend of NO2. The NO2 is important precursor of O3, if the NO2 is underestimated, why dose the performance of the O3 simulation be normal?

4. Why do you choose $4\mu$g m-3 as the threshold to present the different scene?

5. In Figure 5, there are large high values area of NOx and VOCs, but it is corresponding with the low values area of O3, especially in Beijing. What is the reason caused this phenomenon? Though the solar radiation is weak in Jan..

6. In page 225-226, "In addition to the strong emission, this observation should be the main reason for the high mass burden of NOx and VOCs in these regions." What does it mean?

7. In page 254, I don't understand the procedures of the sensitivity tests, if you reduce 30% of VOC emissions or 30% the NOx emissions within the entire model domain, respectively. It should represent the influence of VOC and NOx, respectively. Why is there the variation of the mass concentration of O3 due to the reduction in VOC and NOx emission at the same time?

8. Due to underestimate NO2, whether does it cause the results that "the urban areas and most O3 pollution regions of NCP were mainly dominated by the VOC-sensitive conditions"? And it causes "removal of the transport and power plant sectors could not

effectively reduce the O3 mass burden and even increased the mass burden in high pollution areas, such as southern Beijing, Tianjin, Tangshan, southern Hebei, Jinan, and other parts of Shandong". Because the most important source of NOx is industry, then transportation and power.

Minor points:

1. In page 204, whether the relationship is close, it doesn't only depend on the value of relationship coefficient; it also depends on whether it has passed the significance test.

2. In Figure 4, the time coordinate in Shanghai is not agreement with other cities.

3. In Table 1 and 2, if it is comparisons of hourly data between simulation and observation, why do you calculate the correlation coefficient between daily observation and simulation, rather than hourly data? What's the unit of variables? Why the number of samples in simulation and observation is different?

4. The modeled results show the NOx, why do you compare with NO2?

5. In Figure 2, it is difficult to distinguish the results in Jan. and Jun. except for temperature.

---

## Referee Comment (RC2) · Anonymous Referee #1 · 2 Jun 2018

China is facing serious air pollution with high PM2.5. Recently ozone (O3) becomes the premier pollutant in summer replacing PM2.5. This study investigates this important issue using the regional air quality modeling system RAMS-CMAQ. The ISAM module is used to track the O3 from major pollution regions for the VOC and NOx-sensitive O3. The brute-force method is used to examine the sensitivity of O3 to the reduction of precursor emission from different sectors, which can provide scientific basis for O3 mitigation strategy. This work is in general a solid contribution to understanding of O3 formation and transport at regional scales. I have the following major and minor comments on the manuscript. After the authors address my comments, I would recommend the acceptance of publication.

[Figure]

Major comments:

1. I would like the authors to add some discussions of the novelty of this study. In the introduction, the authors mentioned many previous studies on the similar topics. How does this study differ from previous studies?

2. it looks that the model still has obvious biases (shown in Figures 3 and 4). I would recommend the authors to add some detailed discussions on the potential factors for the model biases: emission, chemistry mechanism, physics, or model grid spacing? Is it possible to add a plot (figure) on VOC (or CO) validation of model results with observations (besides NO2 and O3 in Figures 3 and 4)?

3. The result of regional contributions of NOx- and VOC-senstivie O3 from different regions (Figures 6 and 7) is interesting. Will different regional contributions add up to be 100% at one given location (i.e., local and non-local contributions)? I would suggest to add a table to show the relative contributions to O3 in several regions (e.g., Beijing, Hebei..) from different local and non-local regions. This will give the readers the idea of O3 sources in different regions (local formation versus precursor transport).

4. Please give the reason for the non-linear change of O3: why does O3 increase in many locations when power-plant O3 precursors are removed (Figure 9j)?

Minor comments:

1. Line 66. change "play a role" to "play an important role". 2. Line 77. change "deeply analyzed" to "through analyzed" 3. Line 85, change "severe" to "strict". 4. Line 89. "The amount of surface O3 is expected to continue increasing as the particulate mass loading decreases due to the emission control strategies employed in the NCP". why? can you explain? 5. Line 103. "statistical response surface method". This is not clear. 6. Line 150. "TSSA"? 7. Line 168. "grid distance" to "grid spacing" 8. Line 225. "this observation". not clear. 9. Line 268. "Figure 7f" should be "Figure 8f".
* * *
[Figure]

2018.

---

## Author Comment (AC1) · 29 Jun 2018

Anonymous Referee #1 China is facing serious air pollution with high PM2.5. Recently ozone (O3) becomes the premier pollutant in summer replacing PM2.5. This study investigates this important issue using the regional air quality modeling system RAMS-CMAQ. The ISAM module is used to track the O3 from major pollution regions for the VOC and NOx-sensitive O3. The brute-force method is used to examine the sensitivity of O3 to the reduction of precursor emission from different sectors, which can provide scientific basis for O3 mitigation strategy. This work is in general a solid contribution to understanding of O3 formation and transport at regional scales. I have

the following major and minor comments on the manuscript. After the authors address my comments, I would recommend the acceptance of publication.

Major comments: 1. I would like the authors to add some discussions of the novelty of this study. In the introduction, the authors mentioned many previous studies on the similar topics. How does this study differ from previous studies?

R: Thanks for this comment. It is true that the purpose of this study which tried to investigate the related source contributions and precursor sensitivity features of O3 in NCP was similar with some of the previous researches. However, it can be seen that the method and tools applied in this study should be different from other studies. The basic modeling system RAMS-CMAQ was developed by our group with more than ten years and several processes in the model was modified for improving the accuracy of simulation in China, such as using more precise underlying surface information (Chen et al., JGR, 2018), improving the description of secondary organic aerosols (Li et al., Atmospheric environment, 2017), developing the chemical and physical processes of nitrogen pollutants (Zhang et al., Tellus B, 2007) and dust (Han et al., Aerosol and Air Quality Research, 2012). Therefore, the tools we applied should be unique here. On the other hand, we kindly think that the discussion about the O3-NOx-VOC sensitivity feature was not be using over NCP. Compared with the traditional "Empirical kinetic modeling approach", this method could provide more clearly sensitive features with high temporal resolution. The discussion of the pollution control which released more appropriate sequence of emission reduction was more efficient should be barely mentioned by other studies as well. We added some of the statement about the update of modeling system in Line 134-136, please see if it is OK.

2. it looks that the model still has obvious biases (shown in Figures 3 and 4). I would recommend the authors to add some detailed discussions on the potential factors for the model biases: emission, chemistry mechanism, physics, or model grid spacing? Is it possible to add a plot (figure) on VOC (or CO) validation of model results with observations (besides NO2 and O3 in Figures 3 and 4)?

R: Thanks for this comment. Sorry we did not gave a clearly expression. Here the underestimation generally referred to the model missed some extreme high values from observation, and mainly appeared in January. It can be seen that the mean of modeled mass concentration was very close to the observed data in June as shown by Table 1 and Table 2 (broadly same at Beijing and Jinan), and the research work was generally focused on the situation in June. Therefore, we kindly think that the modeled results were able to be applied for analyzing below. However, we may provide unclear expressions which caused some misunderstanding about the model simulation accuracy, and we have modified the sentences in Line 204-212. Please check if they are OK. On the other hand, we are sorry that the observed data of VOCs was hard to get because it is not the routine monitoring object, and only few specific field campaign measurements had the comprehensive data, but not released in public. In addition, some of the modeled VOC and OC aerosol data by this modeling system have been evaluated in another study (Li et al., Atmospheric environment, 2017), we kindly think that the modeled VOCs result also can be used for analysis.

3. The result of regional contributions of NOx- and VOC-senstivie O3 from different regions (Figures 6 and 7) is interesting. Will different regional contributions add up to be 100% at one given location (i.e., local and non-local contributions)? I would suggest to add a table to show the relative contributions to O3 in several regions (e.g., Beijing, Hebei..) from different local and non-local regions. This will give the readers the idea of O3 sources in different regions (local formation versus precursor transport).

R: Thanks for this comment. Yes, the contributions from all traced sources are equal to the total mass burden of base case, which means the results are 100% conserved. This is an important feature of the ISAM. In addition, we agree that the specific percentage of regional contribution is needed to be shown. Thus, the regional contribution percentage of major regions was added in Table 3, and the related discussion was also added in Line 238-242. Please check if it is OK.

4. Please give the reason for the non-linear change of O3: why does O3 increase in many locations when power-plant O3 precursors are removed (Figure 9j)?

R: Thanks for this comment. The result shown in Figure 9(j) was derived from the brute-force sensitivity tests that could capture the nonlinear effect due to emission reduction. As shown in Figure 8, the regions of O3 increase due to remove of power plant emission were broadly all covered by the "VOC control" type, and generally coincide with the location of strong power plant sources. Thus, the emission of NOx should be saturated and plentiful NOx mass burden would restrain the O3 formation: $O3 + NO \rightarrow NO2 + O2$ In addition, the emission of VOCs from power plant was very small, which means the VOCs mass burden would almost be invariant with reduction of power plant emission. Therefore, in our opinion, the ambient environment would be benefit for the O3 formation when the NOx mass burden decrease due to reduce of power plant emission in the regions shown in Figure 9(j). We added this explanation in Line 297-301, please check if it is OK.

Minor comments: 1. Line 66. change "play a role" to "play an important role". R: Thanks for this comment. We modified the sentence.

2. Line 77. change "deeply analyzed" to "through analyzed" R: Thanks for this comment. We modified the sentence.

3. Line 85, change "severe" to "strict". R: Thanks for this comment. We modified the word.

4. Line 89. "The amount of surface O3 is expected to continue increasing as the particulate mass loading decreases due to the emission control strategies employed in the NCP". why? can you explain? R: Thanks for this comment. Yes, this sentence may lead to misunderstanding, and we have modified the expression here.

5. Line 103. "statistical response surface method". This is not clear. R: Thanks for this comment. We added the explanation here.

6. Line 150. "TSSA"? R: Thanks for this comment. We added the explanation here.

7. Line 168. "grid distance" to "grid spacing" R: Thanks for this comment. We modified this phrase.

8. Line 225. "this observation". not clear. R: Thanks for this comment. We modified this expression.

9. Line 268. "Figure 7f" should be "Figure 8f". R: Thanks for this comment. We modified this word.

Please also note the supplement to this comment:
https://www.atmos-chem-phys-discuss.net/acp-2018-209/acp-2018-209-AC1-supplement.pdf

[Figure]

**Supplement:**

[revised manuscript text omitted]

---

## Author Comment (AC2) · 29 Jun 2018

In the manuscript, the air quality modeling system RAMS-CMAQ (regional atmospheric modeling system-community multiscale air quality), coupled with the ISAM (integrated source apportionment method) module is applied to investigate the O3 regional transport and source contribution features during a heavy O3 pollution episode in June 2015 over NCP. It explores that that the emission sources in Shandong and Hebei was the major contributors to O3 production in the NCP, and it found that the modeling system can provide valuable information for precisely choosing the emission sector combina-

tion to achieve better efficiency. It is meaningful. I recommend the manuscript to be accepted after some minor revisions, and detail some issues below.

Major points: 1. The modeled and observed wind directions were not in good agreement with each other, even in Jan. and Jun. How can you get the result that about 20-30% and 25-40% O3 mass burdens in Beijing and Tianjin were contributed by the emission sources in Shandong and Hebei? Whether should the author compare with the regional atmospheric circulation field?

R: Thanks for this comment. Actually, we kindly think the direct comparison between simulation and observation data should be difficult, especially for the wind direction. In this paper, the observation data of wind direction were obtained from the ground base monitoring sites. These data were significantly impacted by the surrounding surface inevitably. In addition, the time resolutions between measurement (10 min average) and model output (1 hour) were also different. Therefore, the deviation of wind direction should be more significant than other parameters in Figure 2. Here we also present the monthly mean wind field from model and NCEP/NCAR reanalysis (https://www.esrl.noaa.gov/psd/data/composites/hour) in January and June, 2015 (Fig. 1). From the Fig. 1, it can be seen that the wind direction of simulation and reanalysis data were almost same with each other. We also modified the expression in Line 190-192, please check if it is OK.

2. In Figure 4, it seems that there are negative values in modeled hourly mass concentrations of O3 in January, how does this result happen?

R: Thanks for this comment. The mass concentration of O3 was very low (about 10-2 $\mu$g m-3) due to lack of the driving force for the photochemical reaction during nighttime, so that the red line in Figure 4 was superposed with the bottom axis when the value dropped near zero. Here we list a larger version (Fig. 2) of the figure as example, maybe the red line can be see clearly.

3. In Figure 3, the model doesn't perform well in reproducing the observation trend of

NO2. The NO2 is important precursor of O3, if the NO2 is underestimated, why dose the performance of the O3 simulation be normal?

R: Thanks for this comment. Here the underestimation generally referred to the model missed some extreme high values from observation, and mainly appeared in January. As shown in Table 1, it can be seen that the mean of modeled mass concentration was just ∼10 $\mu$g m-3 lower than those of observation at Beijing, Jinan, and Shijiazhuang in January, and broadly same at Beijing, Jinan, and ∼10 $\mu$g m-3 higher at Shijiazhuang and Tianjin in June. Therefore, the underestimation here should not be a systematic error, which may not significantly influence the O3 formation. On the other hand, the modeled NO2 was obviously lower than that of the observation at Tianjin in January (more than 20 $\mu$g m-3). It can be found that this phenomenon caused the performance of O3 simulation was not well at Tianjin: the model somehow overestimated the O3 mass burden in January (The nonlinear relationship between O3 and the precursors). The statement in Line204-212 was not well and may lead to misunderstanding, so we modified it. Please check if it is OK.

4. Why do you choose 4$\mu$g m-3 as the threshold to present the different scene?

R: Thanks for this comment. The "4 $\mu$g m-3" is a typical threshold that recognizes the O3-precursor sensitivity relationship. Several previous studies have used this definition: Sillman et al., 2009; Nie et al., 2014. This definition was developed by Sillman et al., 2002. They applied a composite of 3-D models to simulate the pollutants in several regions. Then, the NOx-VOC sensitivity features were discussed and concluded the threshold for distinguishing various situations. The "5 $\mu$g m-3" for 1-Hour O3 and "4 $\mu$g m-3" for 8-Hour average O3 was found by their studies: S. Sillman and D. He, 2002: Some theoretical results concerning O3-NOx-VOC chemistry and NOx -VOC indicators, Journal of Geophysical Research, 107(D22), 4659, and we added the reference.

5. In Figure 5, there are large high values area of NOx and VOCs, but it is corresponding with the low values area of O3, especially in Beijing. What is the reason caused

this phenomenon? Though the solar radiation is weak in Jan.

R: Thanks for this comment. The low value of O3 mass burden mainly appeared in January. We kindly think that the solar radiation should be an important reason. The chemical reactions of tropospheric O3 formation are a couple of photochemical reactions that derived by the solar radiation. The dramatic diurnal variation of O3 can prove this feature as well. Therefore, the weak radiation should cause less O3 formation potential in winter. On the other hand, most of the atmospheric reactions are competing with each other in atmosphere, so that more NOx and VOCs should tend to format more aerosols if the ambient condition was not benefit for the O3 formation during the winter time. The observation data in Figure 4 also reflected this seasonal variation phenomenon in Beijing and other cities. The average O3 mass concentration was just 10-34 $\mu$g m-3 in winter and reached 69-106 $\mu$g m-3 in summer as shown in Table 2.

6. In page 225-226, "In addition to the strong emission, this observation should be the main reason for the high mass burden of NOx and VOCs in these regions." What does it mean?

R: Thanks for this comment. Sorry, we have modified this sentence, please check if it is OK.

7. In page 254, I don't understand the procedures of the sensitivity tests, if you reduce 30% of VOC emissions or 30% the NOx emissions within the entire model domain, respectively. It should represent the influence of VOC and NOx, respectively. Why is there the variation of the mass concentration of O3 due to the reduction in VOC and NOx emission at the same time?

R: Thanks for this comment. Sorry we may not gave a clear explanation about the sensitivity tests. Here we only conducted two kinds of sensitivity tests: the first one was the sensitivity test that reduced 30% VOC emission, and the second one was the sensitivity test that reduced 30% NOx emission. We did not do these two sensitivity tests at the same time.

8. Due to underestimate NO2, whether does it cause the results that "the urban areas and most O3 pollution regions of NCP were mainly dominated by the VOC-sensitive conditions"? And it causes "removal of the transport and power plant sectors could not effectively reduce the O3 mass burden and even increased the mass burden in high pollution areas, such as southern Beijing, Tianjin, Tangshan, southern Hebei, Jinan, and other parts of Shandong". Because the most important source of NOx is industry, then transportation and power.

R: Thanks for this comment. Sorry we may not gave a clear explanation about the comparison. Actually, the underestimation of NO2 just mean the model missed some extreme high values of the observed NO2 in January. However, here the discussion about the precursor control types and the emission sector contribution features were focused in June, and the simulation results in June were generally well. The mean of modeled NO2 was broadly same as those of the observed one at Beijing, Jinan, and about 10 $\mu$g m-3 higher at Shijiazhuang and Tianjin in June. Therefore, it can be seen that the model performed well in June and the results can be used to discuss the precursor sensitivity and contribution over the model domain. On the other hand, most of the regions with intensive anthropogenic activities in NCP were under VOCs control because the NOx emission was strong. This feature was also coincided with previous studies. We have modified the sentence in Line 255-256.

Minor points: 1. In page 204, whether the relationship is close, it doesn't only depend on the value of relationship coefficient; it also depends on whether it has passed the significance test.

R: Thanks for the comment. Yes, the significance test should be a good method to evaluate the simulation results. On the other hand, we kindly think that the statistic parameters used in this paper could also well reflect the accuracy of simulation results: the mean can reflect whether the magnitude of simulation results was close to the observation; the standard deviation can reflect whether the fluctuation range of simulation results was similar with the observation; the correlation coefficient reflect

the consistency of variation trend between model results and observation data. If the model results perform well in these three aspects, it can be certain that the model reproduced the physical variable reasonably. Yu et al. (2006) has discussed and proved the reasonability of these statistic parameters. Therefore, we kindly think the statistic here used should be appropriate.

2. In Figure 4, the time coordinate in Shanghai is not agreement with other cities.

R: Thanks for the comment. Sorry, we have modified the figure and please check if it is OK.

3. In Table 1 and 2, if it is comparisons of hourly data between simulation and observation, why do you calculate the correlation coefficient between daily observation and simulation, rather than hourly data? What's the unit of variables? Why the number of samples in simulation and observation is different?

R: Thanks for the comment. Actually, the correlation coefficients in Table 1 and Table 2 were calculated from the hourly model and observation data here. The comparison of meteorological parameters were daily mean data. The comparison of NOx and O3 were hourly data.

4. The modeled results show the NOx, why do you compare with NO2?

R: Thanks for the comment. This was mainly because the observation data was NO2. NO2 is one of the routine monitoring pollutants of the Ministry of Environmental Protection of China. The model outputs NO and NO2 simultaneously, so that the NOx can be obtained by sum NO and NO2 directly.

5. In Figure 2, it is difficult to distinguish the results in Jan. and Jun. except for temperature.

R: Sorry, we have modified Figure 2, please check if it is OK.

Please also note the supplement to this comment:

https://www.atmos-chem-phys-discuss.net/acp-2018-209/acp-2018-209-AC2-supplement.pdf

[Figure]

January

[Figure]

[Figure]

June

[Figure]

**Fig. 1.**

[Figure]

**Fig. 2.**